# An Assessment of InP/ZnS as Potential Anti-Cancer Therapy: Quantum Dot Treatment Increases Apoptosis in HeLa Cells

Victoria Davenport [1], Cullen Horstmann [1], Rishi Patel [2], Qihua Wu [2] and Kyoungtae Kim [1,*]

1   Department of Biology, Missouri State University, 901 S National, Springfield, MO 65897, USA;
    Davenport17@live.missouristate.edu (V.D.); Horstmann95@live.missouristate.edu (C.H.)
2   Jordan Valley Innovation Center, Missouri State University, 542 N Boonville Ave, Springfield, MO 65806, USA;
    rjpatel@missouristate.edu (R.P.); qwu@missouristate.edu (Q.W.)
*   Correspondence: kkim@missouristate.edu; Tel.: +1-417-836-5440; Fax: +1-417-836-5126

**Abstract:** InP/ZnS quantum dots (QDs) are an emerging option in QD technologies for uses of fluorescent imaging as well as targeted drug and anticancer therapies based on their customizable properties. In this study we explored effects of InP/ZnS when treated with HeLa cervical cancer cells. We employed XTT viability assays, reactive oxygen species (ROS) analysis, and apoptosis analysis to better understand cytotoxicity extents at different concentrations of InP/ZnS. In addition, we compared the transcriptome profile from the QD-treated HeLa cells with that of untreated HeLa cells to identify changes to the transcriptome in response to the QD. RT-qPCR assay was performed to confirm the findings of transcriptome analysis, and the QD mode of action was illustrated. Our study determined both IC50 concentration of 69 µg/mL and MIC concentration of 167 µg/mL of InP/ZnS. It was observed via XTT assay that cell viability was decreased significantly at the MIC. Production of superoxide, measured by ROS assay with flow cytometry, was decreased, whereas levels of nitrogen radicals increased. Using analysis of apoptosis, we found that induced cell death in the QD-treated samples was shown to be significantly increased when compared to untreated cells. We conclude InP/ZnS QD to decrease cell viability by inducing stress via ROS levels, apoptosis induction, and alteration of transcriptome.

**Keywords:** quantum dot; InP/ZnS; HeLa; viability; apoptosis; transcriptome

## 1. Introduction

HeLa cells are widely known for their use in advancing medical research as well as their controversial origin. From the lab of Dr. George Gey in 1951, cervical cancer cells from an American woman named Henrietta Lacks were cultured starting the HeLa cell line [1]. These cells went on to become the first "immortal" human cells to be used across the world for creating innovative treatments and better understanding the human cancer cell [2].

Many advancements have been made in cancer research since the discovery of HeLa cells and specific interest has been placed on the use of quantum dot (QD) nanoparticles. QDs have fluorescent properties because of their ability to absorb photons, which results in the formation of an electron-hole pair/exciton [3]. When an electron is excited by the bulk valence band of a semiconductor, an electron-hole pair is formed. It is then confined with the nanocrystal until recombined and colored light is emitted [4]. Changing the size of QD varies the emitted color. This makes QDs highly competitive compared to older technologies. The fluorescent quality has attracted interest of many researchers for various uses such as intracellular network labeling, cancer cell marking, radioprotection, increasing radiation efficiency, and detection of pH changes in extracellular environments [3,5,6].

When it comes to imaging technologies, QDs have raised alarm for inducing cellular cytotoxicity. Among these, QDs composed of cadmium (Cd) cores, such as Cd selenide (Se), Cd telluride (Te), and Cd sulfide (S), are the most commonly studied and have become well-known for reducing cell viability [7]. Cd is a transition metal, located in group XII of

the periodic table, and is known for being a human carcinogen. The toxicity has been linked to the release of $Cd^{2+}$ ions that damage the cells it comes into contact with [8]. Shells of zinc sulfide (ZnS) have been added to the Cd QD and have shown to slightly reduce cytotoxicity and highly improve biocompatibility [9].

In efforts to resolve the undesirable side effects, including excess cytotoxicity, in Cd QDs, other core materials have been explored. Included in these efforts are graphene quantum dots (GQDs). GQDs are sought after for their high biocompatibility and low cytotoxicity compared to previous QDs. GQDs have shown great potential for neuroscience advancements and chemotherapy treatment/delivery [10–12].

Another emerging alternative is Indium core QDs. Indium, a metal in group XIII, is combined with phosphide, a $P^{3-}$ ion, and coated with a zinc sulfide shell to become InP/ZnS. Made in the same manner as Cd/ZnS QDs, InP/ZnS is a less commonly investigated QD initially introduced by scientists as a safer alternative to Cd QDs [8]. Currently, InP/ZnS QDs is a relatively newly developing field that lacks enough data to confirm whether the InP QD is more/less toxic compared to Cd. With more research and knowledge about this alternative QD, it can be determined if InP is biocompatible with mammalian cancer cells and deemed safe enough for clinical applications.

Current knowledge on InP QDs is far from complete and more observations will be needed before being suitable for medical practices. The use of QDs as effective treatment against cancerous cells is particularly undiscovered. As found in previous studies, InP/ZnS QDs increased oxidative stress shown by measurements of reactive oxygen species (ROS) as well as inducing apoptosis [13]. In this study, we aimed to observe and expand knowledge on the effects of InP/ZnS QD when introduced to HeLa cells and mice fibroblasts. We hope to better understand if InP is a viable alternative Cd QD and hypothesized InP/ZnS to decrease HeLa cell viability by induction of oxidative stress and cell death. These are measured by XTT cell viability, reactive oxygen species (ROS), apoptosis analyses, and quantification of treated HeLa cell transcriptome compared to the human genome. These components collectively led to physiological pathways that explain how the QD interacts with the cell. While further investigation is still required, the present findings provide promising insight to the use of InP/ZnS QDs as an effective anti-cancer therapy.

## 2. Experimental Section

### 2.1. InP/ZnS Quantum Dots

InP/ZnS QDs, conjugated with carboxylic acid surface ligands (InP/ZnS–COOH), were obtained from NN-Labs (nn-labs.com; Fayetteville, AR, USA) and suspended in water (1 mg/mL). The QDs are capped with ZnS shell surrounding the InP core for stability and to improve biocompatibility. Green InP/ZnS QDs emit at 530 nm $\pm$ 15 nm and are $3.7 \pm 0.5$ nm in diameter.

### 2.2. Cell Culture

Two cell types were observed within the present study: HeLa-S3 cells, authenticated by Genetica Labs of Burlington, NC, and mice fibroblast. The fibroblasts were studied during XTT viability assay to compare the effects of QD treatment in both cancerous and non-cancerous tissues. The protocols for both cell lines were the same. First, cells were taken from $-80$ °C freezer and thawed until reaching 37 °C. Thawed cells were then mixed with 10 mL of Dulbecco's modified Eagle medium (DMEM) media. DMEM was prepared with 10% penicillin and streptomycin antibiotics and 10% fetal bovine serum (FBS) (Mediatech, Inc., Manassas, VA, USA). The cell suspension was then centrifuged at $400\times g$ for 10 min. Afterwards, the supernatant was removed, and the pellet was resuspended in 13 mL of prepared DMEM and plated into a 75 cm$^2$ plate and grown until confluent. The plate was stored in an incubator maintaining 37 °C with an atmosphere of 5% $CO_2$/95% air; media was changed every few days as needed to maintain cell growth.

### 2.3. XTT Viability Assay

An XTT assay was performed separately on both HeLa-S3 and mouse fibroblast cells to compare viability between cancerous and non-cancerous mammalian cells. Protocols and XTT assay kit were obtained from the Biotium manufacturer (www.biotium.com). Using a flat-bottom 96 well plate, HeLa-S3 or mouse fibroblast cells were seeded (7500 cells/well) and incubated at 37 °C for 24 h. On the second day, DMEM was removed and replaced with fresh media (100 μL/well). Designated treatment wells were given 20 μL of InP/ZnS (1 mg/mL) and serially diluted by a factor of 5 to give a concentration range of 0.13 μg/mL to 167 μg/mL. Each concentration was repeated in triplicate and the experiment was repeated twice. Dimethyl sulfoxide (DMSO) was used as a positive control in separate wells at three different concentrations: 5%, 10%, and 15%. The positive control was also done in triplicate. Treated cells were incubated for another 24 h. After incubation, an activated XTT solution was mixed in 1:200 ratio of XTT solution and XTT Activation Reagent (PMS), respectively. Each well was treated with 25 μL of the prepared XTT solution for 7 h at 37 °C. This time frame allows for adequate formation of dye for absorbance measurements. The formazan dye that formed in the wells was quantified with BioTek ELx880 Absorbance Microplate Reader (www.biotek.com; Winooski, VT, USA) (absorbance measured at wavelength 450 nm). The A 450 nm absorbance value was subtracted by that at A 630 nm to accurately measure the amount of dye after 7 h of incubation.

### 2.4. Calculation of $IC_{50}$ Value

To calculate the concentration at which 50% of cells were inhibited ($IC_{50}$), AAT Bioquest website (https://www.aatbio.com/tools/ic50-calculator) was used. Data from our XXT cell viability assay were uploaded and the $IC_{50}$ value was calculated by computer to be 69 μg/mL. The equation used by computer can be seen in Equation (1) below. The equation generates a regression model by using the maximum and minimum values from the uploaded data set; the Hill coefficient represents ligand cooperativity. The $IC_{50}$ value models the inflection point of the sigmoid function. The calculated $IC_{50}$ value was used in all consecutive experiments as the main treatment concentration.

Equation (1): Inhibition concentration 50% ($IC_{50}$) equation form used by computer on AAT Bioquest website (https://www.aatbio.com/tools/ic50-calculator) to calculate the $IC_{50}$ value. This value was used in following experiments as the tested treatment concentration.

$$Y = Min + \frac{Max - Min}{1 + \left(\frac{X}{IC_{50}}\right)^{\text{Hill coefficient}}} \tag{1}$$

### 2.5. Reactive Oxygen Species Assay

An assay to measure the reactive oxygen species (ROS) was performed over a three-day period as described before [14–16]. Initially, cells were cultured and counted. Into two separate 24-well plates (one for dihydroethidium (DHE) and dihydrorhodamine 123 (DHR)), 50,000 cells were seeded into each well and placed into an incubator for 24 h. The next day, warmed DMEM media was removed and replaced in each well of both plates. In labeled columns, two different concentrations of InP/ZnS were applied equally to both plates: 69 μg/mL (designated $IC_{50}$ value), and 167 μg/mL (shown to significantly decrease cell viability via XTT assay, Figure 4). After treatment, the plates were put back into the incubator for another 24 h. On the third day, the cells were detached from the well for harvesting. After removing the DMEM, wells were washed twice with 500 μL 1X phosphate-buffered saline (PBS) (OmniPur®). Once washed, 250 μL of trypsin + EDTA was added to each well and incubated for 15 min. To neutralize the trypsin after incubation, 250 μL of warmed DMEM was added, and the contents of each well were transferred to labeled 2 mL microcentrifuge tubes to be put into the centrifuge for 10 min at $400 \times g$. During centrifugation, two ROS indicator solutions (DHE and DHR) (Biotium, San Francisco, CA, USA) were prepared by mixing 0.5 mg of indicator with 1 mL DMSO. The supernatant of each centrifuged tube was removed, and the pellet was resuspended in 10 μL of the corresponding

indicator and 990 µL of 1XPBS. Tubes were then covered in foil to be incubated at 37 °C for 30 min before being analyzed by Attune NxT Flow Cytometer (Thermo Fisher Scientific, Waltham, MA, USA). Samples dyed with DHE were measured at 518/606 nm and those with DHR at 507/536 nm.

### 2.6. Apoptosis Assay

Before beginning the apoptosis assay, 10X Annexin V Binding buffer was prepared by mixing concentrations of 0.1M HEPES, 1.5 M sodium chloride (NaCl), 25 mM calcium chloride (CaCl$_2$), and molecular grade sterile water. For the experiment, the 10X Annexin V Binding buffer was diluted with 1XPBS to be a 1X solution. The experiment was initiated by seeding 50,000 HeLa cells into each well of a 24-well plate. After incubating for 24 h, original DMEM media was replaced and treatment was applied (both 69 µg/mL and 167 µg/mL InP/ZnS). The plate was placed into an incubator for another 24 h. On the third day, wells were washed two times with 1XPBS and treated with 250 µL non-EDTA Trypsin. After incubating for 20 min, wells were given 250 µL fresh DMEM media. The samples were transferred to microcentrifuge tubes and spun for 10 min at 2000× *g*, after which the supernatant was removed. The pellets of untreated and negative control samples were resuspended in 500 µL 1XPBS. Positive controls and treated samples were resuspended in 100 µL 1X Annexin V Binding buffer, 5 µL Annexin V-APC, and 5 µL propidium iodide (PI) (BD Biosciences, San Jose, CA, USA). After incubating in the dark for 30 min, another 400 µL of 1X binding buffer was added to each sample. Samples were then observed using Attune Nxt Flow cytometer (Thermo Fisher Scientific, Waltham, MA) based on excitation properties of the dyes. Annexin V-APC emits at 660 nm with a red laser, and PI emits at 617 nm with a blue laser.

### 2.7. Total RNA Extraction and cDNA Conversion

Total RNA extraction was performed on HeLa cells treated with 100 µg/mL InP/ZnS. To begin, 750,000 cells were seeded into each well of a 6-well plate and incubated for 24 h. Using Invitrogen TRIzol protocols, total RNA was extracted and then resuspended in 30 µL of nuclease-free water. Samples were then quantified using a Qubit 3.0 Fluorometer at 280 nm OD. With TruSeq® Stranded mRNA LT Sample Preparation Kit (Illumina, San Diego, CA, USA), mRNA was then isolated and used to synthesize 6 cDNA samples (3 non-treated and 3 InP/ZnS treated) using Verso Reverse Transcriptase (Thermo Scientific protocol). Each sample was ligated by a different adaptor and amplified 15 times in T100TM Thermal Cycler (www.BIO-RAD.com).

### 2.8. Transcriptome Analysis

To determine significantly altered genes, cDNA sequences of treated and non-treated samples were obtained from the University of Kansas Medical Genome Centre. The cDNA samples were sequenced with 100 nucleotide sequences of each end by using the Illumina Hiseq 2500 sequencing system. Next, the sequence from each sample was transferred to Basepair Tech website (www.basepairtech.com). Once uploaded, pipeline RNA-seq and total expression count was conducted on the sequences. Results were aligned with the human reference genome (hg19) via differential expression (DESeq2) to produce a list of effected genes. The list of genes was then transferred to GOrilla, where Gene Ontology (GO) terms were obtained and grouped by significance and function using Microsoft Excel.

### 2.9. RT-qPCR

In order to confirm the changes in the transcriptome of HeLa cells treated with green InP/ZnS QDs (530 nm), real-time reverse transcriptase polymerase chain reaction was performed. About 1 ng of RNA was used to synthesize cDNA (three control and three samples treated with InP/ZnS QD) using the Verso cDNA conversion kit (Thermo Fisher Scientific, Waltham, MA, USA). After quantifying cDNA with Qubit 3.0 fluorometer (Thermo Fischer Scientific, Waltham, MA), DNA primers were designed for target genes based on RNAseq

results: ID2 was significantly downregulated in InP/ZnS presence; PTGIR was upregulated and CCT2 was not significantly altered by QD presence (deemed to be housekeeping gene). To verify their function a primer efficiency test was done. cDNA samples were diluted by a factor of 2 and amplified with PCR (GoTaq qPCR kit, Promega, Madison, WI, USA). Primer efficiency values were found to be within 1.6–2.3. After confirming efficiency, 30 ng of cDNA from each sample, along with a blank control, were used to amplify the three target genes with GoTaq qPCR master mix protocol (Promega). The wells were mixed thoroughly by pipette and centrifuge, and the plate was put into QuantStudio 6 Pro instrument (Thermo Fisher Scientific, Waltham, MA, USA) for PCR amplification. Changes were detected in comparison to the housekeeping gene (CCT2) using the Pfaffl method [14,15,17]; the gene expression ratio of each target gene was calculated based on E (RT-qPCR efficiencies) and Cq value deviation from the control (CCT2 expression levels).

### 2.10. Chemophysical Properties

A chemical and physical properties analysis of the commercially purchased QDs was conducted using scanning transmission electron microscopy (STEM), energy dispersive x-ray spectroscopy (EDAX), ultraviolet-visible (UV–VIS) spectroscopy, and dynamic light scattering (DLS). The stock (INPW530) QD material was diluted to a concentration of 100 µg/mL and was used for each proceeding study.

A JEOL 7900F scanning electron microscope (SEM) with a scanning transmission electron microcopy (STEM) detector was used to image the QDs for qualitative dimensional analysis. The diluted QD dispersion was drop cast onto holy carbon film TEM grids and allowed to air dry in a desiccator box prior to imaging. Elemental composition of the InP/ZnS QDs was measured using a Bruker Quantax energy dispersive x-ray spectrometer (EDAX) attached to the SEM. A thick film of the InP/ZnS QDs was deposited using a drop cast method onto a silicon substrate prior to analysis.

Dynamic light scattering (DLS) technique was used to characterize the hydrodynamic size of the InP/ZnS QDs which were diluted in water and cell media. The diluted samples were tested with a Malvern Panalytical Zetasizer® Ultra with laser wavelength of 632 nm and scattering angle of 173°. The UV-Vis absorption spectrum was recorded with a Shimadzu UV3600 UV-Vis-NIR Spectrometer with scan range of 200 nm to 800 nm and a step size of 0.5 nm.

### 2.11. Statistical Analysis

GraphPad Prism 6.0 was used to statistically analyze data of all experiments. One-way ANOVA analysis and Dunnett's multiple comparisons were done to visualize the variance in control and treatment groups. On all graphs, each sample is represented by an average of three replicates and have error bars representing standard deviation. Statistically significant data are represented on graphs as * $p < 0.05$, ** $p < 0.01$, *** $p < 0.001$, **** $p < 0.0001$.

## 3. Results

### 3.1. Chemophysical Properties of Green InP/ZnS

The drop casted QDs on holy carbon TEM grids shows their distribution as both individual as well agglomerated particles. This is due to the drying process of the QDs from the solution. The STEM images (Figure 1) provide a qualitative approximation of the InP/ZnS QD particle sizes in agreement with that of the datasheet values ($3.7 \pm 0.5$ nm) [18]. EDAX was performed to verify the composition as well as determine whether any potential cytotoxicity is not due to elemental composition. The EDAX spectra (Figure 2) verifies that the elemental composition of the QD sample is composed of InP/ZnS.

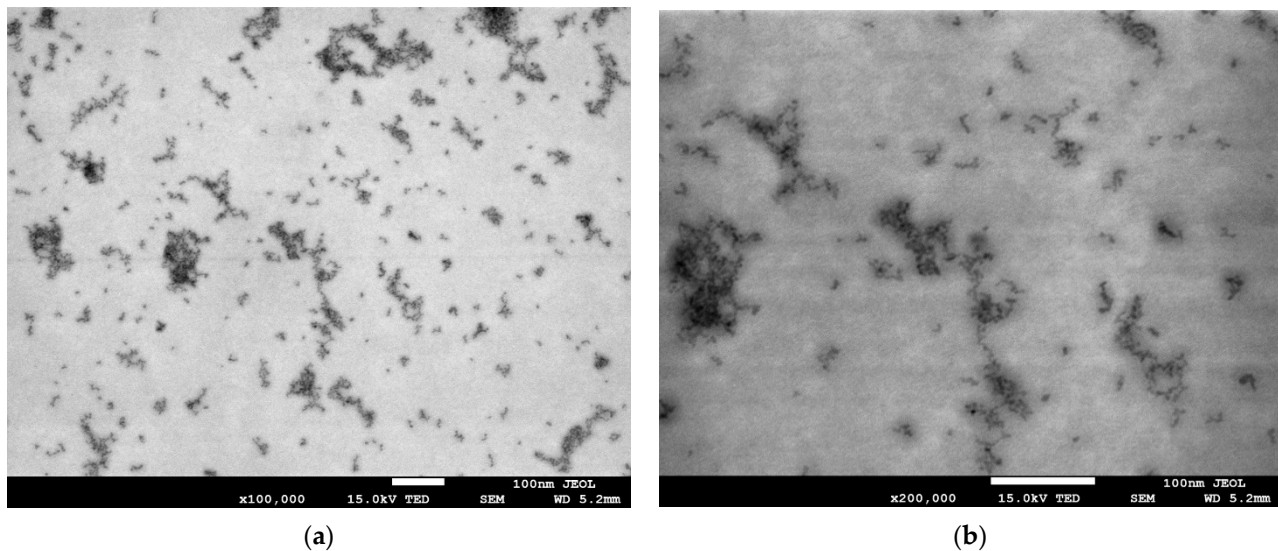

**Figure 1.** STEM images: Drop casted InP/ZnS (530 nm) quantum dots (QDs) on holy carbon TEM grids at low (**a**) and high (**b**) magnifications.

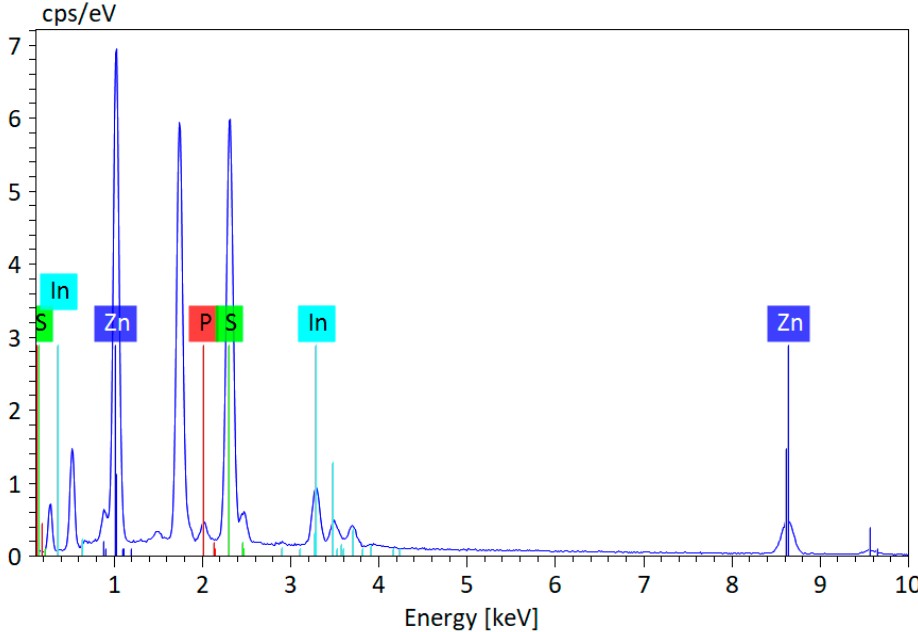

**Figure 2.** EDAX spectra of InP/ZnS (530nm) QD to verify elemental composition.

The DLS study was conducted on QDs diluted in both water and cell media. The DLS analysis (Figure 3) shows a monodisperse peak with a hydrodynamic diameter of 72.3 nm (QDs diluted in water) and 89.7 nm (QDs diluted in cell media). The results indicate that there might be formation of ligands shells (carboxylic acid ligands) and possibly some aggregation of the QDs dispersed in both solutions, where slightly more agglomeration is evident in the cell media. The UV–VIS data of InP/ZnS QDs showed no absorption at or above wavelengths of 650 nm (Supplementary Materials, Figure S1a). The first absorption peak for InP/ZnS was around 470 nm (Supplementary Materials, Figure S1b), in agreement with the datasheet [19]. The thick film of InP/ZnS QDs that were deposited onto a silicon substrate for EDAX analysis was excited using a UV source at 365 nm, which shows emission at 530 nm/Green (Supplementary Materials, Figure S2). Photoluminescence spectra can be found in the datasheet where the quantum yield >10% for this QD (INPW-530) material [18,19].

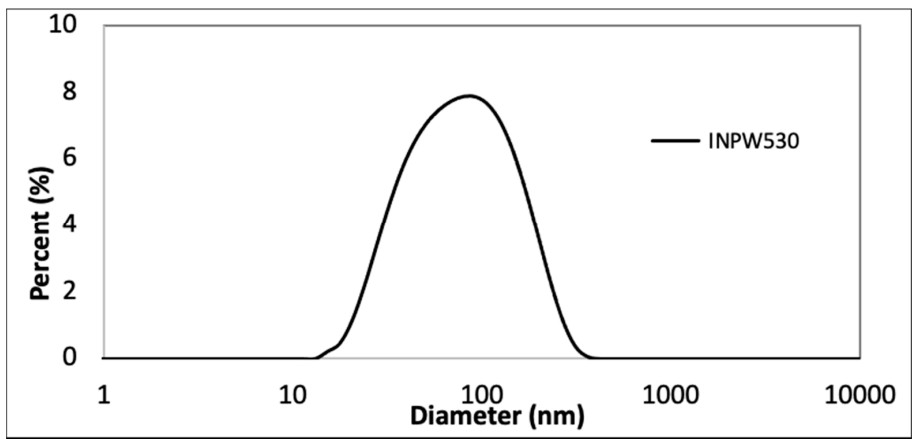

**Figure 3.** DLS intensity size distribution: QDs diluted to 100 μg/mL in both water and cell media shows an effect hydrodynamic size of 72.3 nm and 89.7 nm, respectively.

### 3.2. Effect on Cell Viability when Treated with InP/ZnS QDs

To examine the toxicity of InP/ZnS QD when treated to HeLa cells, an XTT assay was performed. Cells were treated with concentrations ranging from 0.13 μg/mL to 167 μg/mL, which were serially diluted by a factor of 6× (Figure 4a). After 7 h of incubation with XTT activation reagent, HeLa cell viability, correlated to measured absorbance levels, was reduced by approximately 32% when treated with 167 μg/mL InP/ZnS compared to the non-treated control. At this concentration, samples averaged an absorbance level of 0.23, which was deemed statistically significant by one-way ANOVA test (Figure 4a). All other concentrations, including the control, averaged absorption levels between 0.31- and 0.33. Based on these results, 167 μg/mL was used as the minimum inhibition concentration for further experiments. A positive control was also performed using DMSO as the treatment and is shown to have lower absorbance levels (and therefore decreased cell viability) when in the presence of DMSO (concentration ranging from 5–20%) compared to the non-treated control.

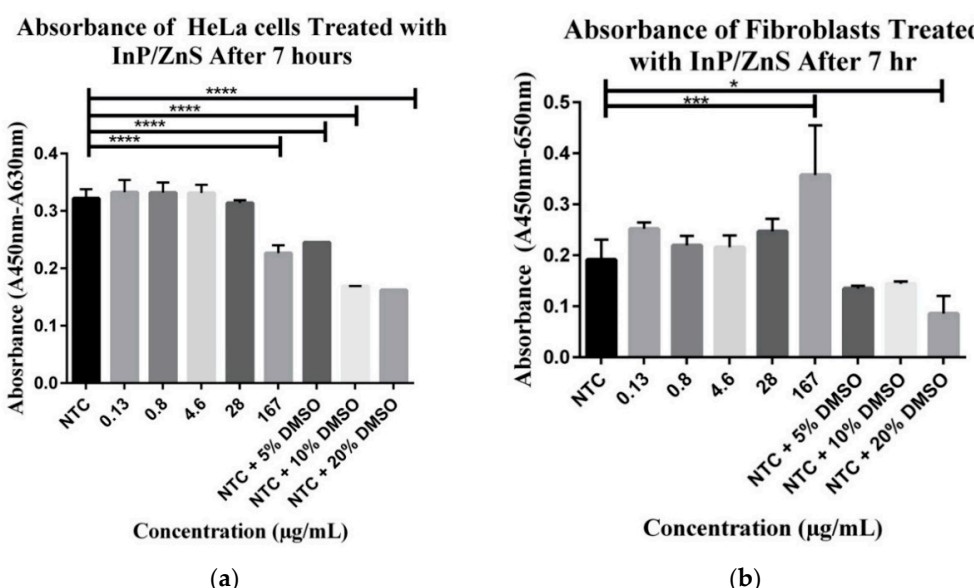

(**a**)                    (**b**)

**Figure 4.** Effects of InP/ZnS QD on cell viability measured by XTT reagents. (**a**) HeLa cells show minor reduction in cell viability shown through 28 μg/mL and significant reduction at 167 μg/mL. Positive control with DMSO treatment shows statistically significant reduction in viability at higher percentages of DMSO. (**b**) Fibroblast cells have no significant decrease at all InP/ZnS concentration treatments. Statistically significant results are indicated based on *p*-values: * $p < 0.05$, *** $p < 0.001$, **** $p < 0.0001$.

Mice fibroblast cells were tested in the same manner to observe the effect of InP/ZnS treatment in non-cancerous cells. Interestingly, there was no decrease in cell viability compared to the non-treated control at all concentration levels (Figure 4b). Fibroblast viability actually resulted in a slight increase when treated with InP/ZnS QDs. The same as with HeLa cells, treatment concentrations ranged from 0.13 µg/mL to 167 µg/mL; at the highest concentration, absorbance was increased compared to the control (0.19 non-treated control vs. 0.36 absorbance at 167 µg/mL). Differing from HeLa cells, increased viability in mice fibroblast does not seem to be concentration dependent. This result is specifically important when exploring QD use for clinical therapies. Further studies are needed to deem InP/ZnS QDs effective in cancerous cells but impotent to surrounding healthy tissue.

### 3.3. Increased Production of Peroxynitrite Radicals in InP/ZnS Presence

The production of reactive oxygen species in the form of free radicals is a known reaction of cellular metabolism that, when increased, plays a large role in disturbing the survival of the cell in stressful environments. We tested for ROS production in cells treated with 69 µg/mL and 167 µg/mL InP/ZnS. Samples were dyed with dihydroethidium (DHE) to measure the production of superoxide radicals, and dihydrorhodamine (DHR) to measure the production of peroxynitrite (Figure 5a,b). Fluorescent peaks of each sample were observed with flow cytometer and gates were used to measure the percentage of ROS generated (Figure 5c–h). The three samples of each concentration were averaged to visualize the differences between them (Figure 5a,b). In observance of superoxide radicals using DHE there was significant decrease in ROS production compared to the dyed, non-treated control (the dyed control averaged 79.18% ROS production). Both concentrations of InP/ZnS treatment yielded similar percentages of change; 69 µg/mL InP/ZnS yielded an average ROS production of 56.34%, and 167 µg/mL InP/ZnS yielded 51.89% (Figure 5a). Contrasting from DHE, the non-treated control dyed with DHR (averaging 57.85% ROS production) increased production of peroxynitrite (Figure 5b). Again, levels of increased ROS were very similar in both concentrations of InP/ZnS treatment. Samples treated with 69 µg/mL InP/ZnS averaged 79.28% ROS production, while those treated with 167 µg/mL InP/ZnS averaged 78.13% ROS production. As superoxide radicals are formed in the mitochondria, they can collide with nitric oxide to form peroxynitrite; this causes cellular dysfunction and cell death [20]. This pathway could help explain the observed decrease in ROS dyed with DHE. As peroxynitrite radical formation is increasing, the superoxide radicals that it is derived from are then depleted.

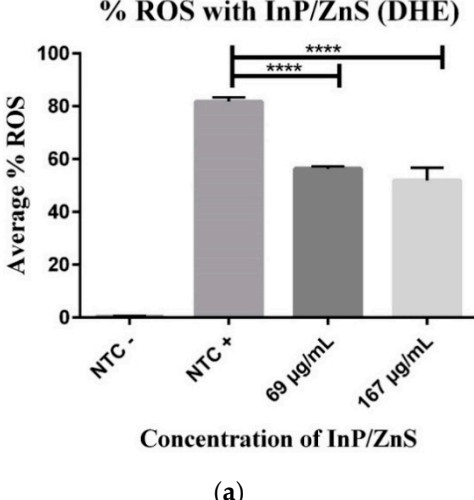

(a)

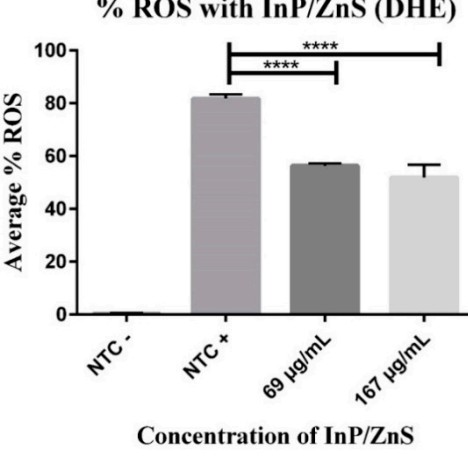

(b)

**Figure 5.** *Cont.*

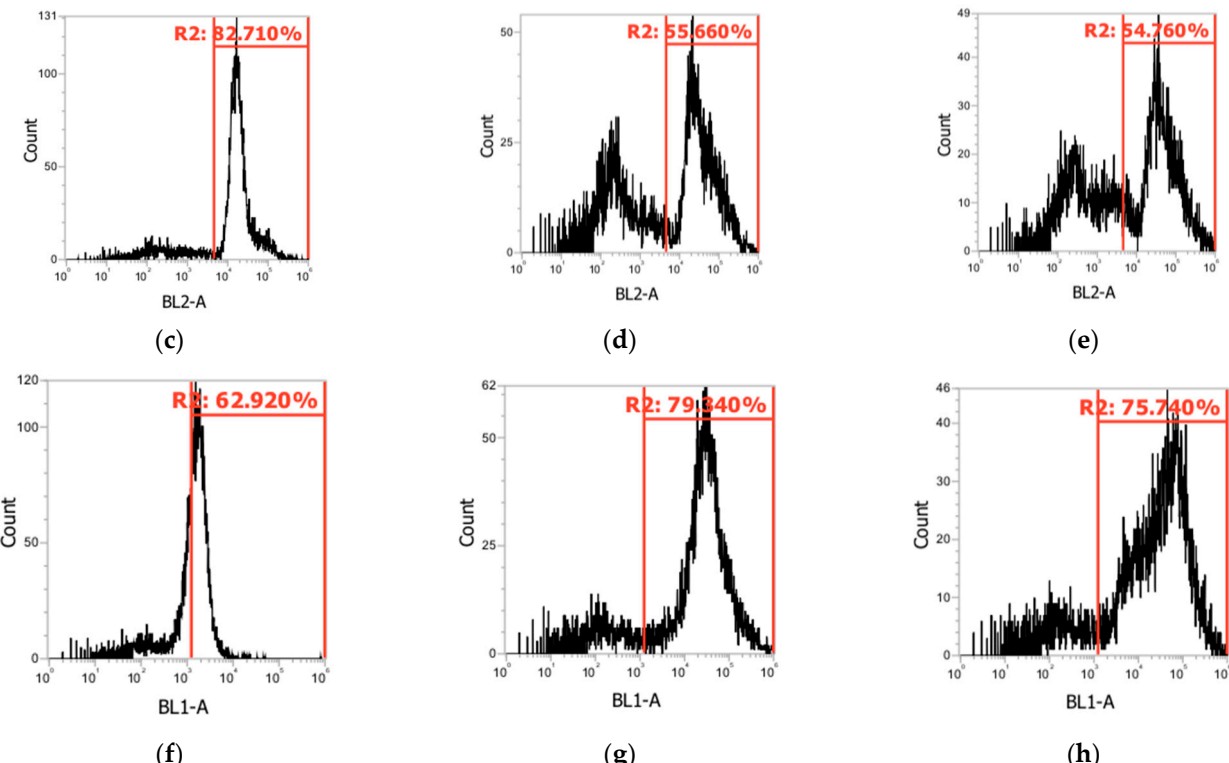

**Figure 5.** ROS measurements with both DHE and DHR at varied InP/ZnS QD concentrations. (**a**,**b**) Bar graphs comparing average percentages of measured ROS when dyed with DHE and DHR, respectively. (**c**–**e**) Flow cytometer peaks taken from ROS assay of samples dyed with DHE. A second larger peak represents ROS production; (**c**) dyed, untreated control; (**d**) dyed sample treated with 69 µg/mL InP/ZnS; (**e**) sample treated with 167 µg/mL InP/ZnS; (**f**–**h**) peaks of samples dyed with DHR; (**f**) dyed, untreated control; (**g**) dyed sample treated with 69 µg/mL InP/ZnS; (**h**) dyed sample treated with 167 µg/mL InP/ZnS. Statistically significant results are indicated based on *p*-values: **** $p < 0.0001$.

### 3.4. Increased Late Apoptosis with InP/ZnS Treatment

Increased levels of ROS induce stress and cause damage to multiple intracellular components that often lead to induction of apoptosis [21]. The understanding of apoptotic-like programmed cell death is important to tumor suppression mechanisms and advancements in anti-cancer drugs [22]. We tested for an induction of apoptosis in cells treated with 69 µg/mL and 167 µg/mL of InP/ZnS QDs. Our results were divided into early and late apoptosis, which were observed with gated quadrants using flow cytometer (Figure 6). Early apoptosis involves the activation of multiple signal cascades and can be measured by the presence of phosphatidylserine (PS) in the outer leaflet of the plasma membrane (detected with annexin V); late apoptosis represents the fragmentation of DNA [23]. In both treatment groups, induced apoptosis was decreased in early phase and increased in late phase. The dyed control for early apoptosis (Figure 6a,c) averaged 27.7%, while that for late apoptosis (Figure 6b,d) averaged 0.69%. There was not a significant difference in the two concentrations of treatment, indicating the IC$_{50}$ (69 µg/mL InP/ZnS) is a sufficient dose to change levels of apoptosis. At 69 µg/mL InP/ZnS, early apoptosis averaged 15.24%, a 45% decrease from the control group. At 167 µg/mL InP/ZnS, samples averaged 19.48% early apoptosis, a 29.7% difference decrease compared to NTC. As for late apoptosis, 69 µg/mL InP/ZnS averaged 31.28%, nearly 4417% increase from the control group. 167 µg/mL InP/ZnS averaged 23.67%, a 3317% increase.

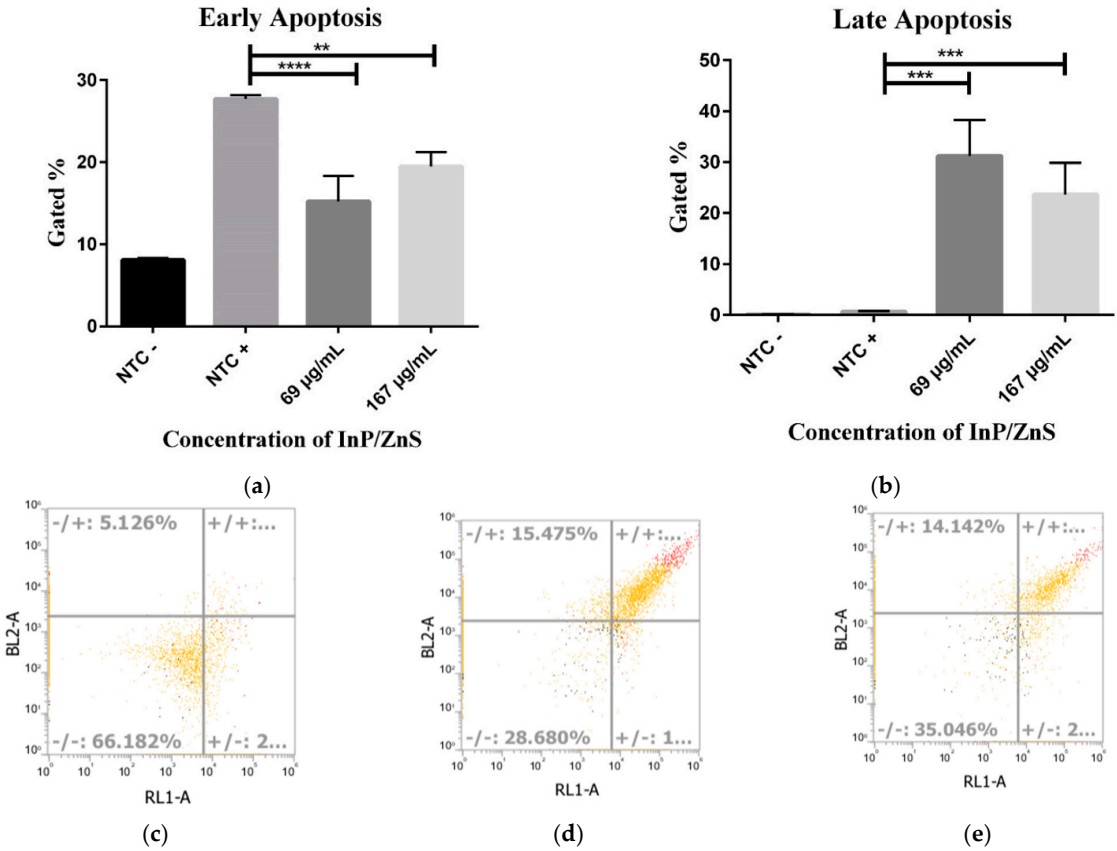

**Figure 6.** Levels of early and late apoptosis after InP/ZnS treatment. (**a**,**b**) Bar graph illustrating percent of apoptosis recorded by a flow cytometer in samples treated with varying concentrations of InP/ZnS. (**c**–**e**) Flow cytometer images of gated apoptosis levels. Each quadrant represents a scheme 69. μg/mL of InP/ZnS, and (**e**) represents 167 μg/mL of InP/ZnS. Error bars indicate standard deviation of samples. Statistically significant results are indicated based on *p*-values: ** *p* < 0.01, *** *p* < 0.001, **** *p* < 0.0001.

### 3.5. Altered Genome with InP/ZnS Treatment

RNA-seq results aligned with the human reference genome showed 16.4k genes differentially expressed. A comprehensive list of these genes, along with their Log2Fold change and *p*-values can be found in the Supplementary Materials. The data were filtered for *p*-value < 0.05, which resulted in 620 downregulated genes and 1129 upregulated genes. The 1129 upregulated genes were sorted by function and yielded strong activation of developmental processes (such as cell differentiation, tissue and nervous system development, and morphogenesis) and signal transduction (including stimulus responses, G protein-coupled receptor signaling, and inflammatory responses) (Figure 7a). Specific genes involved with developmental processes include *LGALS9*, *ZSWIM4*, and *IFITM1*; those involved with signal transduction are *RFNG*, *MEIS3*, and *SYN1*. The same process was performed with the 620 downregulated genes. This yielded major disruption (over 300 altered genes) of metabolic processes and biosynthetic process regulations (Figure 7b). The list of genes related to metabolic processes included *NCBP2*, *DPM1*, and *BCAS2*; biosynthetic process genes consisted of *PSMC6*, *PSMD12*, and *PSMD14*. A second filter was then applied to select genes with a log2 fold change greater than 1. This resulted in 69 downregulated and 408 upregulated with *p* < 0.05 and log2 fold change > 1. The functions of the highly downregulated genes were investigated to show significant disruption to cell growth regulation, protein quality controls, and mitochondrial function. Notable genes in this category are *PDK4*, *ID2*, *SRRM4*, and *MMP13*. Highly upregulated genes commonly coded for transmembrane proteins to alter proliferation and, cellular transport, and organization. Significant genes for upregulation include *TIMP1*, *ZSWIM4*, and *LGALS9*.

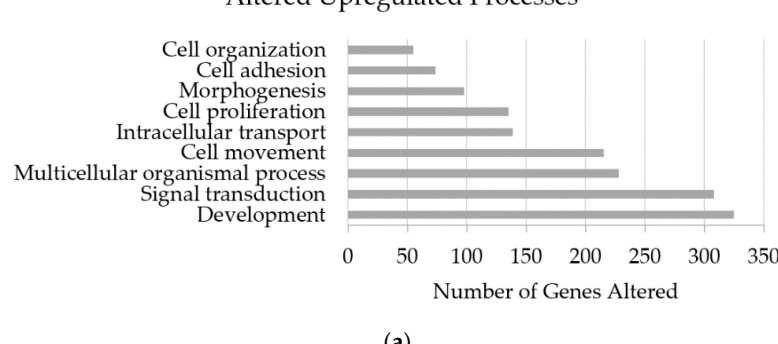

(**a**)

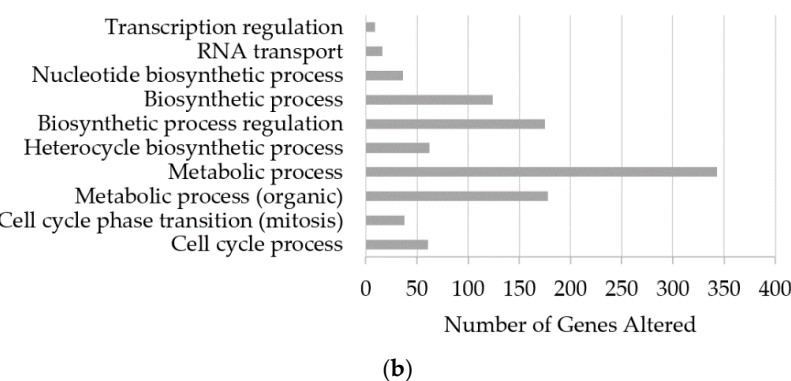

(**b**)

**Figure 7.** Altered up and down regulated processes: Bar graph representing number of genes upregulated (**a**) and downregulated (**b**), based on having *p*-value < 0.05. Genes are grouped by common processes.

### 3.6. RNAseq Confirmation with RT-qPCR

Using the Pfaffl method [17], gene expression ratios of one up regulated (*PTGIR*), one down regulated (*ID2*), and a housekeeping control gene (*CCT2*) were measured from real-time RT-qPCR data. Ratios equaling <1 and >1 represent down and up regulation, respectively. After treatment with 100 μg/mL InP/ZnS, a triplicate sample with *PTGIR* averaged 3.86. In triplicate samples with *ID2*, 0.574 was seen. These values were compared to that of the housekeeping control, *CCT2*, set to a ratio of 1 (Figure 8). The ratios revealed with RT-qPCR are consistent with gene expression changes seen with RNAseq: *PTGIR* up regulated with fold change of 5.13 and *ID2* down regulated with fold change of −3.50.

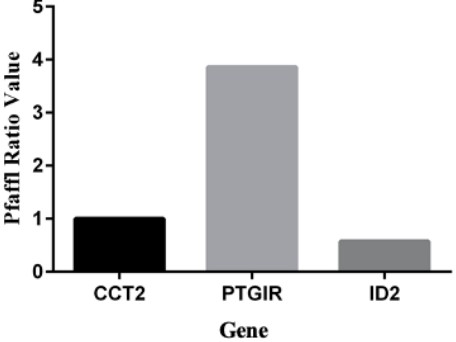

**Figure 8.** RT-qPCR gene expression ratios for RNAseq validation. Genes *PTGIR* (up regulated), *ID2* (down regulated) and *CCT2* (housekeeping control) were chosen from RNAseq data to be analyzed by RT-qPCR with the Pfaffl equation. Y axis measures *Pfaffl* gene expression ratio where <1 and >1 show up and down regulation, respectively.

## 4. Discussion

QDs have potential to advance medical techniques such as molecular imaging and anti-cancer treatment. InP/ZnS QDs have been proposed as a safer alternative to Cd-based QDs but have not been previously investigated enough to understand how they alter cellular homeostasis, function, and transcriptome. This study aims to fill the gap in this knowledge. Through a series of experiments, we hypothesize pathways in which InP/ZnS QDs travel through HeLa cells and the effects they have. We found increased levels of oxidative stress and apoptosis, which supports our previously stated hypotheses that suspect toxicity is associated with QD treatment. Investigation of the transcriptome via RNAseq and RT-qPCR offer evidence of transcriptome alteration and are explained with cellular models mapping these pathways. Our findings are significant for understanding more about InP/ZnS QD effects, but further investigation is required to understand if they are a viable technology for medical treatment.

### 4.1. Changes to HeLa Cell Viability, ROS, and Apoptosis Levels with InP/ZnS Treatment

When observing cell viability after 24 h of InP/ZnS treatment, 167 μg/mL InP/ZnS significantly decreased cell survival by 32%. Among the current data, the consensus on InP/ZnS effect on cell viability seems to vary by cell type and duration of treatment. For example, Chen T et al. observed varying decrease in viability after 24 and 48 h of treatment to human lung cancer cell (HCC-15) and alveolar type II epithelial cell (RLE-6TN) (treatment concentrations ranging 80–160 μg/mL InP/ZnS). REL-6TN cells with 160 μg/mL were 80% viable after 24 h of treatment, and only 40% viable after 48 h [8]. In a similar study, treatment for 24 h at 80 μg/mL and below did not significantly decrease cell viability in bone marrow-derived macrophages of mice [13]. Collectively, these findings suggest that HeLa cells are susceptible to InP/ZnS-related damage and are shown in the current study to be reduced in viability by 32% compared to NTC. Further investigation with varying treatment concentrations and time periods could aid in better understanding this comparison.

The influences of QDs on ROS production, specifically InP/ZnS QDs, has been previously reported many times [24–27]. ROS consist of many metabolites that induce stress within the cell including superoxide, hydrogen peroxide ($H_2O_2$) and peroxynitrite. Each of the metabolites mentioned are measured with a different fluorescent probe: DHE, DCFH-DA, and DHR, respectively. Superoxide can merge with nitric oxide to form peroxynitrite, or superoxide dismutase to make $H_2O_2$. Focusing on DHE and DHR, we observed cellular production of superoxide radicals decreased, while peroxynitrite levels increased. This contrasts from data by Ayupova et al. that reports increased production of superoxide radials in RAW 264.7 murine "macrophage-like" cells. The authors note that lysosomal activation in HeLa cells could be correlated to the increase of ROS production, something that would need further investigation within our own study [28]. Observing $H_2O_2$ production, T. Chen et al. reported significant increases in ROS in two different lung-derived cell lines [8]. Collectively, it seems that ROS is increased in some aspect across varying cell lines but there is not yet a conclusion to which metabolite is most influential in creating cellular stress.

Increased oxidative stress by ROS has been linked to inducting apoptosis as well. T. Chen et al. observed increased levels of ROS and apoptosis in lung-derived cell lines. This is consistent with an observed increase of late apoptosis in our study, however T. Chen et al. reported significant increase of apoptosis at a much lower concentration of InP/ZnS QDs (20 μg/mL) [8]. At 28 μg/mL InP/ZnS, we did not find cell viability to be significantly changed (Figure 4); again, variation in InP/ZnS toxicity is documented between cell types and presents an area of study to be expanded in the future.

### 4.2. Upregulated Gene Processes Induce Apoptosis and Inhibit Metastasis

By analyzing up regulated genes from InP/ZnS QDs treatment, genes controlling metastatic inhibition and increased cell aggregation were overexpressed. *TIMP1* is an in-

hibitor of matrix metalloproteinase (MMP) and involved in apoptotic pathways of synovial fibroblasts. Prior studies of Dumortier et al. have shown *MMP13* role in breast tumorigenesis. In their findings, the inhibition of MMP was linked to regulating cell migration and apoptosis [29]. Thus, *TIMP1* gene up regulation, as seen in our analysis, could promote increased control over metastasizing cells when treated with InP/ZnS QDs.

Also found in apoptotic pathways of fibroblasts is *ITGA10*: an integrin protein involved in cell adhesion and surface signaling. When Saftencu et.al. observed papillary thyroid cancer RNA, *ITGA10*, and other biological adhesion genes suggested a prognostic impact in linking gene expression to patient survival [30]. We hypothesize that the up regulation of adhesion genes could play an important role in tumorigenesis by impacting tumor cell aggregation and metastasis.

Evidence of apoptosis induction was seen again by *LGALS9* (Figure 9a). This gene is responsible for suppressing T-cell proliferation, inducing T-cell apoptosis, and associated with angiogenesis and cytokine signaling. Activation results in autocrine release of Gal9, and is linked to inducing apoptosis [31]. This is supported in our apoptosis tests (Figure 6), where a statistically significant increase of late apoptosis was observed in InP/ZnS-treated cells compared to NTC.

Increased *ZC3H12A* expression also points to apoptosis. *ZC3H12A*, a transcription factor, is proposed to induce apoptotic gene expression and regulate inflammation/stress responses. Studied in colorectal cancer patients, T. Chen et al. found lower expression levels of *ZC3H12A* in more aggressive tumor cells [32]. Thus, increasing *ZC3H12A* expression could be effective in making cancerous tissue less aggressive. Going forward, targeted therapies could explore uses of this gene in inducing tumor cell death.

We also report up regulation of *ZSWIM4*, potentially involved in chromatic organization. We suspect disruption of this gene could lead to disruption of *ZC3H12A* as well (Figure 9a). Our results suggest InP/ZnS QDs induces stress on the cell, which increases *ZSWIM4* and *ZC3H12A* expression and induces apoptosis.

Along with *ZC3H12A*, *LIF* and *LGALS9* upregulation has been associated with collateral damage via the immune system. Leukemia inhibitory factor (*LIF*) is a member of interleukin 6 pro-inflammatory cytokine family and is responsible for inhibiting cell growth and differentiation. In their study with elephant cells, Vazquez et al. reported upregulation of *LIF* results in DNA damage and leads to apoptosis; the authors propose this to be linked to the evolution of cancer resistance in organisms with long life spans [33]. Immune function was also observed by Holderried et al. by investigation of T cell-mediated tumor evasion. Galactin-9, a ligand made by *LGALS9*, was upregulated in colorectal tumors and hypothesized to be a therapeutic target for immune-mediated anti-cancer therapies [31]. Collectively, we suggest cellular stress could activate the immune system in the presence of InP/ZnS QDs and induce apoptosis in tumor cells (Figure 9a).

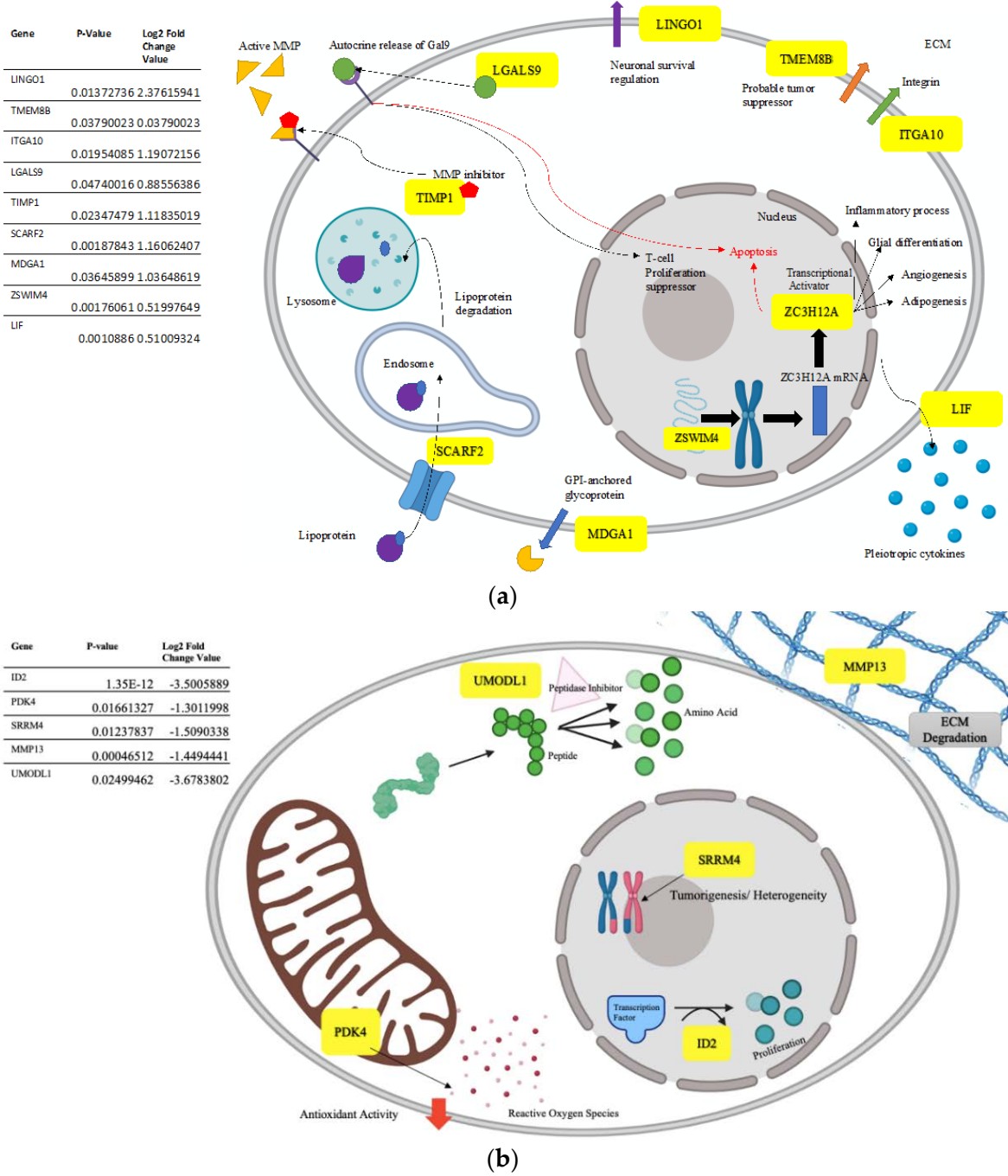

**Figure 9.** Visualization of altered processes in the presence of InP/ZnS. Upregulated processes (**a**) observe evidence of metastatic inhibition, apoptosis due to autocrine release, and cytokine release. Downregulated processes (**b**) show pathways involving proliferation inhibition, increased ROS production, and varying levels of degradation. To the left of each model is a chart depicting each gene involved, along with the corresponding Log2 Fold Change value and *p*-value found with transcriptome analysis.

### 4.3. Downregulated Processes also Prevent Tumor Growth and Spread

Earlier, up regulation of *LGALS9* and *ZC3H12A* was linked to apoptosis. Similarly, down regulation of *ID2* could inhibit cell proliferation. *ID2* (Figure 9b) has been investigated as an important activator in tumor progression in glioma and breast tissues [34–36]. Gene targeting could further be investigated for preventing tumor growth.

Alternative splicing in mRNA creates recombinant and diverse proteomes within a cell and increases tumor aggressiveness; in previous studies, a common regulator and potential

therapeutic target of this process is *SRRM4* [37,38]. Presently, *SRRM4* levels were inhibited by InP/ZnS QD treatment, possibly showing an ability to alter the *SRRM4* communication pathway (Figure 9b). Signaling mechanisms of *SRRM4* have been proposed to show an ability to drive tumor progression, aiding in future development of resistance therapies [39].

Similar to *LGALS9*, downregulation of *PDK4* suggests potential to induce apoptosis. A member of the pyruvate dehydrogenase kinase (PDK) family, PDKs function to regulate carbohydrate metabolism in mammals by catalyzing oxidative decarboxylation of pyruvate [40]. Overexpression of PDKs is linked to tumor formation from antioxidants that increase cancer cell metabolism and survival [40,41]. From this, we suspect the observed PDK inhibition (Figure 9b) to possibly explain the decrease in superoxide radical production (Figure 5a) as a result of controlled oxidative phosphorylation and glucose metabolism [40,42,43]. Thus, we hope our data will contribute to the potential research of PDK regulation as an anti-cancer treatment.

*UMODL1* is another gene that was down regulated (Figure 9b). This gene is associated with immune and female reproductive systems and peptidase inhibition, however data explaining its role in cancer is sparse. Available studies have linked *UMODL1* to lung cancer metastasis [44] and ovarian degradation when upregulated [45]. Originally, we had suspected a decrease in *UMODL1* expression to show tissue degradation by excess peptide breakdown. However, this is contradicting to the findings of W. Wang in which they link ovarian degradation to an increase of *UMODL1* expression [45]. While ovarian and cervical tissues are closely related in the body, it is important to note that gene response can vary among organ tissues. Linkage to excessive tissue breakdown pose a negative side effect to InP/ZnS QD treatment in HeLa cells, however an expansion in testing is necessary to further understand *UMODL1's* influence in cervical cancer.

We mentioned above that *TIMP1* expression inhibits that of MMPs. We support this with our RNAseq results that yield increased *TIMP1* and decreased *MMP13* expressions (Figure 9b). *MMP13* is an MMP transcription factor in the collagenase family. We suspect upregulation of *TIMP1* to have a role in this. *MMP13* has been found to be overexpressed in breast cancer tissues and contributes to the breakdown of ECM collagen to promote metastasis [29,46]. Stemming from this, we hypothesize that inhibition of MMPs can prevent tumor metastasis via inhibition of ECM degradation.

Our findings support InP/ZnS treatment to have potential in controlling tumor growth by suppressing proliferation and metastasis. The observed decrease in cell viability is a concerning component of QDs and should be considered going forward. We acknowledge the limitations within this study and suggest subsequent experiments with HeLa cells and InP/ZnS QDs before recommending them to be a safe, reliable option for medical applications and beyond. While the current study shows a broad range of treatment concentrations, it lacks small intervals between each other. The large gaps in treatment concentrations limit data that would pinpoint the concentration of InP/ZnS that significantly decreases cell viability. Future studies may focus of smaller ranges of treatment to understand this point further. With diversified studies, better understanding of QD-inflicted toxicity will progress the potential contribution to clinical treatment and technology.

## 5. Conclusions

The toxicity of QDs has raised concerns against their uses in new medical technologies. Being proposed as a safer alternative to Cd-based QDs, InP QDs were studied for their effects on viability, reactive oxygen species, apoptosis, and the transcriptome of HeLa cells. At concentrations of 69 and 167 μg/mL InP/ZnS cell viability was depleted, while ROS and apoptosis levels were elevated. The transcriptome had many alterations in presence of the InP/ZnS QD that can be linked to preventing proliferation and metastasis, as well as inducing apoptosis and cytokine release. There is potential for InP/ZnS QDs to be an effective treatment against tumor progression, but its toxic side effects should be considered. Going forward, studies should investigate different treatment concentrations and lengths of exposure to broaden data and understanding.

**Supplementary Materials:** The following are available online at https://www.mdpi.com/2624-8 45X/2/1/2/s1, Figure S1: UV-Vis absorption spectra of INPW530, Figure S2: UV (365 nm) light excitation of InP/ZnS QDs 530 nm.

**Author Contributions:** Conceptualization, V.D., C.H., and K.K.; methodology, V.D., C.H., R.P., and Q.W.; software, V.D.; validation, V.D. and K.K.; formal analysis, V.D., C.H., R.P., and Q.W.; investigation, V.D., C.H., R.P., and Q.W.; resources, K.K.; data curation, V.D. and C.H.; writing—original draft preparation, V.D.; writing—review and editing, V.D., K.K., R.P., and Q.W.; visualization, V.D.; supervision, K.K.; project administration, K.K.; funding acquisition, K.K. All authors have read and agreed to the published version of the manuscript.

**Funding:** This research was funded by the US Army Engineer Research and Development Center-Environmental Laboratory through the Environmental Quality and Technology Program, Contract No. W912HZ19C-0048.

**Acknowledgments:** The author would like to acknowledge B. Hens and J. Reel for their support in executing this project.

**Conflicts of Interest:** The authors declare no conflict of interest. The funders had no role in the design of the study; in the collection, analyses, or interpretation of data; in the writing of the manuscript, or in the decision to publish the results.

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
