# Peer review of "An Assessment of InP/ZnS as Potential Anti-Cancer Therapy: Quantum Dot Treatment Increases Apoptosis in HeLa Cells"

_jnt, doi:10.3390/jnt2010002_

Round 1
Reviewer 1 Report
The manuscript “An assessment of InP/ZnS as potential anti-cancer therapy: Quantum dot treatment induces stress on HeLa cells” explores the potential of quantum dots (QDs) in anticancer therapy, by investigating metabolic activity of HeLa cells and mouse fibroblasts, reactive oxygen species production and QDs-induced apoptosis of HeLa cells, along with transcriptome analysis of HeLa. Taken together, results emerging from this study contribute to deepening knowledge in the rising QDs-based therapeutic field, not in the diagnostic one. I do recommend this article for publication after minor revision.
Authors should discuss XTT data in Figure 6. They observed a significant increase in cell viability of fibroblasts after the treatment with 167 µg/mL of QDs. However, this augmented viability was not concentration dependent. An explanation of this result should be provided.
Figure 6. I suggest keeping the same direction for concentrations of DMSO (from 20% to 5% or from 5% to 20%) to simplify the take-home message.
Toxicity of Cd-based QDs is mentioned in the paper. However, a rising strategy that allows to use biocompatible QDs is to exploit graphene quantum dots (GQDs), which do not exert toxic effects on cells. Several studies on GQDs have demonstrated their ability to act as sensitizers or carriers for chemotherapy and phototherapy. A brief discussion of these pieces of evidence could be included.
(https://doi.org/10.3390/ijms21176301, https://doi.org/10.1021/ja312221g, https://doi.org/10.3390/ijms21103712 ).
Authors should expand section 3.6.
Author Response
Reviewer 1:
The manuscript “An assessment of InP/ZnS as potential anti-cancer therapy: Quantum dot treatment induces stress on HeLa cells” explores the potential of quantum dots (QDs) in anticancer therapy, by investigating metabolic activity of HeLa cells and mouse fibroblasts, reactive oxygen species production and QDs-induced apoptosis of HeLa cells, along with transcriptome analysis of HeLa. Taken together, results emerging from this study contribute to deepening knowledge in the rising QDs-based therapeutic field, not in the diagnostic one. I do recommend this article for publication after minor revision.
We thank you greatly for your time and effort in editing our manuscript. Your insight has been applied to the updated manuscript and we feel the writing has been improved from your suggestions.
Authors should discuss XTT data in Figure 6. They observed a significant increase in cell viability of fibroblasts after the treatment with 167 µg/mL of QDs. However, this augmented viability was not concentration dependent. An explanation of this result should be provided.
-
- Section 3.2 has been expanded to better discuss this result. We are unsure why this result was yielded but suggest further studies for future explanation.
- Figure 6. I suggest keeping the same direction for concentrations of DMSO (from 20% to 5% or from 5% to 20%) to simplify the take-home message.
- Figure 6 A has been adjusted to match the axis titles in 6b.
- Toxicity of Cd-based QDs is mentioned in the paper. However, a rising strategy that allows to use biocompatible QDs is to exploit graphene quantum dots (GQDs), which do not exert toxic effects on cells. Several studies on GQDs have demonstrated their ability to act as sensitizers or carriers for chemotherapy and phototherapy. A brief discussion of these pieces of evidence could be included.
- (https://doi.org/10.3390/ijms21176301, https://doi.org/10.1021/ja312221g, https://doi.org/10.3390/ijms21103712 ).
- The introduction has been revised to reflect this comment. Now seen in Line 55-60, graphene quantum dots are mentioned along with InP/ZnS QDs.
- Authors should expand section 3.6.
- Thank you for the suggestion. To improve section 3.6, we have added greater detail to the listed genes found in our RNAseq to be differentially expressed. We have also added a complete excel table of all resulting genes from RNAseq, which will be placed in the supplementary materials.

Reviewer 2 Report
The authors assessed InP/ZnS quantum dots as anticancer agent. XTT assay, ROS and RT-qPCR results are presented.
Why numbers are presented in Keywords? What means fluoresced? Is there really releasing of calcium in oxidation state +3? Indium phosphide is not a basic metal. Suppliers of chemicals are not reported. Producers of some instruments are not mentioned or fully given. Shortcuts for time should be used. Meaning of symbols in equation is not reported. Explanation of shortcuts should be given first time they appear. There are too many figures in the manuscript. I see no reason to present characterization of quantum dots as it is given in datasheet. It can be moved to supplementary. STEM images without scale are illustrations only. Why 7 h was chosen for cell viability? Why mice fibroblast cells?
Author Response
Reviewer 2:
The authors assessed InP/ZnS quantum dots as anticancer agent. XTT assay, ROS and RT-qPCR results are presented.
Thank you for your efforts in reviewing our manuscript. We hope each of your comments has been satisfied and feel the manuscript has been greatly improved by your inputs.
Why numbers are presented in Keywords?
-
- Originally, they meant to function as a list, however the numbers have been removed for clarity. Thank you.
- What means fluoresced?
- At line 42, “color is fluoresced” has been changed to “colored light is emitted” for better clarity. Thanks.
- Is there really releasing of calcium in oxidation state +3?
- This was a mistake by the author that has now been changed to the correct term “Cd2+”
- Indium phosphide is not a basic metal.
- The elemental explanation of Indium phosphide has been altered for better accuracy; see line 64 “Indium, a metal in group XIII, is combined with phosphide, a P3- ion, and coated with a Zink Sulfide shell to become InP/ZnS”.
- Suppliers of chemicals are not reported.
- Producers of some instruments are not mentioned or fully given.
- To the best of our knowledge, all chemicals and instrument suppliers/producers are now included in full.
- Shortcuts for time should be used.
- This has been applied throughout the manuscript for all mentions of time.
- Meaning of symbols in equation is not reported.
In section 2.4 the following sentence was added for my best effort to better explain the components of the equation:
"The equation used by computer can be seen in Equation 1 below. The equation generates a regression model by using the maximum and minimum values from the uploaded data set; the Hill coefficient represents ligand cooperativity. The IC50 value models the inflection point of the sigmoid function."
- Explanation of shortcuts should be given first time they appear.
- Abbreviated element or solution names have now been explained the first time they appear throughout the manuscript.
- There are too many figures in the manuscript. I see no reason to present characterization of quantum dots as it is given in datasheet. It can be moved to supplementary.
- We appreciate your suggestion here. To accommodate, figures 4 and 5 have been moved to supplementary materials. A similar manuscript from our lab has previously been published in this journal with the same volume of figures and we feel that they are essential to proper characterization of our findings. Thank you for the feedback.
- STEM images without scale are illustrations only.
- The STEM images have been corrected to have a scale.
- Why 7 h was chosen for cell viability?
- Cell viability was measured after 7 hours to allow adequate time for reactions to take place and absorbance to be measured. This has been expanded within the writing at line 122 for better clarity.
- Why mice fibroblast cells?
- To better clarify this, line 99-100 has been added into section 2.2 to explain the use of fibroblast cells. It is also explained in 3.2.

Reviewer 3 Report
The authors report the theranostic affect of InP/Zn QDs on Hela cancer cells. Resulst obtained show that the level of nitrogen radicals significantly increased in QDs-treated samples and that the dots decrease the viability of the cells. Some of the results may be of interest but major corrections are required. From my opinion, the manuscript suffers from the modest characterization of the dots used.
- line 40-41 : clarify the sentence. A valence band electron is not excited by a semiconductor.
- the authors must highlight the novelty of their work compared to previous reports.
- line 50 : Ca3+ (?) should be changed into Cd2+.
- the production of ROS from QDs is well documented in the literature (J. Hazard. Mater. 2016, 304, 532-542). References related to the production of ROS by InP/ZnS QDs should also be added (Angew. Chem. Int. Ed 2019, 58, 11414-11418; Nanoscale 2011, 3, 2552-2559; ACS Appl. Mater. Interfaces 2019, 11, 12367-12378).
- along the whole manuscript, results must be better discussed in the context of literature.
- Figure 1: STEM images are of modest quality. Scale bars and a size distribution should also be added.
- line 192 : green-emitting InP/ZnS
- paragraph 3.1.: displace the informations related to the suppliers of the equipments to the experimental section.
- paragraph 3.1.: the whole paragraph should be revised. 1) DLS results are surprising and the authors indicate that the dots are agregated (or poorly dispersible in water). The dispersion of the QDs in the biological buffer should also be evaluated. 2) The PL emission spectrum and the PL QY of the dots should also be provided.
- from my opinion, figure 4b and figure 5 can be removed.
- lines 257-267 : from my opinion, the discussion should be revised. ROS have a very short lifetime (microsecond scale for superoxide radicals). Can these radicals relay travel through the electron transport chain and combine with nitric oxide as indicated by the authors ?
- the interaction of InP/ZnS with Hela cells must be evaluted, for example by fluorescence microscopy.
Author Response
Reviewer 3:
The authors report the theranostic affect of InP/Zn QDs on Hela cancer cells. Results obtained show that the level of nitrogen radicals significantly increased in QDs-treated samples and that the dots decrease the viability of the cells. Some of the results may be of interest but major corrections are required. From my opinion, the manuscript suffers from the modest characterization of the dots used.
We thank you for your time and effort in editing our work. Major corrections have been made to follow your suggestions. The characterization has been edited and results better explained. We believe the manuscript to be much improved after following your comments.
line 40-41 : clarify the sentence. A valence band electron is not excited by a semiconductor.
-
- The citation for this sentence was revisited and the writing was corrected. The new sentence, seen at line 40-41 should now reflect better accuracy. Thank you for the feedback.
- the authors must highlight the novelty of their work compared to previous reports.
- Lines 79-84 have been expanded to address this comment. The novelty is specifically noted.
- line 50 : Ca3+ (?) should be changed into Cd2+.
- This mistake has been corrected.
- the production of ROS from QDs is well documented in the literature (J. Hazard. Mater. 2016, 304, 532-542). References related to the production of ROS by InP/ZnS QDs should also be added (Angew. Chem. Int. Ed 2019, 58, 11414-11418; Nanoscale 2011, 3, 2552-2559; ACS Appl. Mater. Interfaces 2019, 11, 12367-12378).
- The provided citations have been added to section 4.1 for better support to the findings of the current study.
- along the whole manuscript, results must be better discussed in the context of literature.
- We have addressed this recommendation by adding 6 new literature citations to the manuscript. Additionally, the results of section 4.2 have been discussed more thoroughly within the context of previous literature. Thank you for the perspective.
- Figure 1: STEM images are of modest quality. Scale bars and a size distribution should also be added.
- Scale and sizing details have been added to these images to improve their quality.
- line 192 : green-emitting InP/ZnS
- This has been added to line 287. We also added this detail to line 96 in section 2.1 for better explanation of the color of QDs used.
- paragraph 3.1.: displace the informations related to the suppliers of the equipments to the experimental section.
An additional methods/experimental section has been added to reflect this, now seen as section 2.10 Chemophysical properties. Thank you for the suggestion.
- paragraph 3.1.: the whole paragraph should be revised. 1) DLS results are surprising and the authors indicate that the dots are agregated (or poorly dispersible in water). The dispersion of the QDs in the biological buffer should also be evaluated. 2) The PL emission spectrum and the PL QY of the dots should also be provided.
This paragraph has been revised. We now include two additional citations in reference to DLS data (references 18 and 19). We hope the referenced data sheets to provide PL QY data. We appreciate the insight.
- from my opinion, figure 4b and figure 5 can be removed.
- To compensate for the large number of figures, 4 and 5 have been moved to supplementary materials and any relevant writing/formatting has been updated to reflect the change.
- lines 257-267 : from my opinion, the discussion should be revised. ROS have a very short lifetime (microsecond scale for superoxide radicals). Can these radicals relay travel through the electron transport chain and combine with nitric oxide as indicated by the authors ?
- Revisions to this can now be seen in lines 342-344 in which it is explained more clearly that superoxide collides with nitric oxide within the mitochondria, but not the electron transport chain specifically. The message of this paragraph should now reflect better accuracy. Thank you for the suggestion.
- the interaction of InP/ZnS with Hela cells must be evaluated, for example by fluorescence microscopy.
- Thank you very much for the suggestion. Respectfully, we believe that this is outside of the project scope for this manuscript however future plans in our lab include a manuscript dedicated to evaluating this.

Round 2
Reviewer 2 Report
Thank you for improvements.
Author Response
This letter is to follow a second revision of our manuscript An assessment of InP/ZnS as potential anti-cancer therapy: Quantum dot treatment induces stress on HeLa cells. The time and effort given by the reviewers is greatly appreciated for improving our manuscript for submission. We have carefully reviewed each comment and put great care into fixing the writing for maximal quality.
After evaluating the manuscript, the reviewers believe the findings of our study are relevant to the aim of the Journal of Nanotheranostics. Our manuscript expands knowledge on quantum dot-based therapeutics and provides data on the effects within mammalian cells.
Upon inspection, the reviewers specifically requested the language be altered to improve the writing overall. To address this concern, we have polished and updated the writing to improve clarity and flow. This was done throughout the manuscript in each section as seen fit. We believe our edits to greatly improve the writing and justify its consideration for publishment.
Revisions can be seen on the updated manuscript with track changes. Detailed responses are seen below in blue italicized font. We hope our edits to satisfy the requests of the reviewers and once again thank you for the opportunity to improve our manuscript for further submission.

Reviewer 3 Report
Most of the scientific changes asked by the reviewers were made by the authors. However, some major changes are still required to improve the manuscript. The authors must carefully check the manuscript, especially the corrections made and improve the language. For example, 1) line 321: QDs are composed of In, P, Zn and S, 2) line 328 : DLS is not a spectrum but an analysis, 3) line 375 : "excitation" should be changed into "emission", 4) line 399 : "when treated with InP/ZnS QD treatment (?), ...
The manuscript contains many errors that make it difficult to read and understand.
Author Response
This letter is to follow a second revision of our manuscript An assessment of InP/ZnS as potential anti-cancer therapy: Quantum dot treatment induces stress on HeLa cells. The time and effort given by the reviewers is greatly appreciated for improving our manuscript for submission. We have carefully reviewed each comment and put great care into fixing the writing for maximal quality.
After evaluating the manuscript, the reviewers believe the findings of our study are relevant to the aim of the Journal of Nanotheranostics. Our manuscript expands knowledge on quantum dot-based therapeutics and provides data on the effects within mammalian cells.
Upon inspection, the reviewers specifically requested the language be altered to improve the writing overall. To address this concern, we have polished and updated the writing to improve clarity and flow. This was done throughout the manuscript in each section as seen fit. We believe our edits to greatly improve the writing and justify its consideration for publishment.
Revisions can be seen on the updated manuscript with track changes. Detailed responses are seen below in blue italicized font. We hope our edits to satisfy the requests of the reviewers and once again thank you for the opportunity to improve our manuscript for further submission.
Reviewer 3 comment:
Most of the scientific changes asked by the reviewers were made by the authors. However, some major changes are still required to improve the manuscript. The authors must carefully check the manuscript, especially the corrections made and improve the language. The manuscript contains many errors that make it difficult to read and understand.
- line 321: QDs are composed of In, P, Zn and S
- This line has been altered to read “InP/ZnS” for better accuracy and flow. Thank you.
- line 328 : DLS is not a spectrum but an analysis
- We have replaced “spectrum” with “analysis” to improve the language. Thank you.
- line 375 : "excitation" should be changed into "emission"
- This correction has been changed to reflect the expertise of the reviewer. Thank you for the suggestion.
- line 399 : "when treated with InP/ZnS QD treatment (?)
- The redundancy has been fixed in this sentence to make it easier to understand. Thank you for pointing it out.
We again thank you for the opportunity to further improve our manuscript. The time and experience of the reviewers is very valuable to us.

Round 3
Reviewer 3 Report
Most of the corrections suggested by the reviewers were made. The following minor comments should be considered by the authors : - clarify the title : An assessment of InP/ZnS quantum dots... - line 40, clarify the sentence : when a valence electron is excited to the conduction band... - line 97 : InP/ZnS QDs capped with carboxylate ligands were obtained... - line 277 : green-emitting InP/ZnS QDs
Author Response
The reviewers requested minor revisions to be made to our manuscript. We have diligently addressed each comment to improve our manuscript. We believe our edits to greatly improve the writing and justify its consideration for publishment.
Revisions can be seen on the updated manuscript with track changes. Detailed responses are seen below in blue italicized font. We hope our edits fulfill the requests of the reviewers and once again thank you for the opportunity to improve our manuscript for further submission.
Reviewer 3 Comments:
Most of the corrections suggested by the reviewers were made. The following minor comments should be considered by the authors:
- clarify the title: An assessment of InP/ZnS quantum dots...
- Thank you for the feedback. We have changed the title to read “...Quantum dot treatment increases apoptosis in HeLa cells.” We believe this to be more impactful of a title and to emphasize the results we found in relevance to anti-cancer therapy.
- line 40, clarify the sentence: when a valence electron is excited to the conduction band...
- The citation for this sentence has been revisited and the writing adjusted. The changes can be seen at line 40 and we believe it to accurately reflect the information from the citation (doi: 1177/0192623307310950). Thank you for the feedback.
- line 97: InP/ZnS QDs capped with carboxylate ligands were obtained...
- This sentence was originally meant to detail the surface ligand on the QD. We have altered the sentence to specifically say “surface ligand” and added an abbreviation “InP/ZnS-COOH” to better represent the full composure of QD. Thank you for the perspective.
- line 277: green-emitting InP/ZnS QDs
- Here we had aimed to describe the wavelength of the QDs used. For better clarity the sentence has been altered to read “green InP/ZnS QDs (530nm)”, found at line 183-184. We believe this to better explain the characterization of the QDs. Thank you.
We thank you again for allowing us to improve our manuscript.
